# Quality of Life and Job Loss during the COVID-19 Pandemic: Mediation by Hopelessness and Moderation by Trait Emotional Intelligence

**DOI:** 10.3390/ijerph19052756

**Published:** 2022-02-27

**Authors:** Federica Andrei, Giacomo Mancini, Francesca Agostini, Maria Stella Epifanio, Marco Andrea Piombo, Martina Riolo, Vittoria Spicuzza, Erica Neri, Rosa Lo Baido, Sabina La Grutta, Elena Trombini

**Affiliations:** 1Department of Psychology “Renzo Canestrari”, Alma Mater Studiorum, University of Bologna, 40127 Bologna, Italy; f.agostini@unibo.it (F.A.); marcoandrea.piombo2@unibo.it (M.A.P.); erica.neri4@unibo.it (E.N.); elena.trombini@unibo.it (E.T.); 2Department of Education Studies “Giovanni Maria Bertin”, Alma Mater Studiorum, University of Bologna, 40126 Bologna, Italy; giacomo.mancini7@unibo.it; 3Department of Psychology, Educational Science and Human Movement, University of Palermo, 90128 Palermo, Italy; mariastella.epifanio@unipa.it (M.S.E.); martina.riolo@unipa.it (M.R.); vittoriaspicuzza@hotmail.it (V.S.); sabina.lagrutta@unipa.it (S.L.G.); 4Department of Biomedicine, Neuroscience and Advanced Diagnostics (Bi.N.D.), University of Palermo, 90129 Palermo, Italy; rosa.lobaido@unipa.it

**Keywords:** COVID-19 pandemic, quality of life, trait emotional intelligence, hopelessness, job loss, TEIQue, WHOQOL, BHS

## Abstract

This study contributes to the knowledge on the effects of the COVID-19 pandemic by examining a moderated mediation model in which the impact of job loss over quality of life (QoL) is mediated by hopelessness and moderated by trait emotional intelligence (trait EI). Data were collected from a large nationally representative Italian sample of adult workers (*N* = 1610), who completed a series of anonymous online questionnaires. Total, direct and indirect effects were estimated through bootstrapped mediated moderation analyses providing 95% bias corrected bootstrap confidence intervals. After controlling for the effects of gender and age range, job loss was found to be negatively associated with QoL, and hopelessness partially mediated such relationship. These relationships were in turn moderated by trait EI. Our study suggests that trait EI levels act as protective factor for a good QoL, mitigating the impact of both job loss and hopelessness over QoL levels during the COVID-19 pandemic. Identifying psychological protective and/or risk factors for a better QoL is crucial for the development of interventions aimed at reducing the emotional impact of the pandemic and of its negative real-life consequences.

## 1. Introduction

The health emergency caused by the COVID-19 pandemic and the consequent restrictive measures issued by the governments have severely tested the ability of individuals to cope with stressful events. Many aspects of people’s lives (emotional, relational, work, social, economic, recreational) have been affected by the pandemic, so much so that current research highlights the impact that the COVID-19 emergency is having on the perception of people’s quality of life (QoL), as well as on their daily psychological and emotional experiences (e.g., [1,2,3,4]).

At the same time, the repercussions from the pandemic have significantly hit the global economy, including an increase in the unemployment rate worldwide [5], especially among young adults, women [6] and in countries that are being hardest affected by the pandemic itself (Italy, Spain, and the United Kingdom [7]). These countries are more likely to suffer the worst employment implications because their labour markets were more vulnerable before the crisis, featured by high unemployment and precarious work [7]. Hence, although the COVID-19 pandemic can be certainly considered a unique stressor on its own, not everyone has been affected by it to the same extent. In fact, people who underwent additional external stressors, such as job loss and subsequent economic hardship, may be classified as being more at risk to develop negative consequences on their health and psychological wellbeing. For instance, current literature suggests that people who had lost their job during the pandemic reported higher symptoms of depression, anxiety and stress, and lower positive mental health compared to individuals who kept working [8]. Researchers have pointed out that these effects can be long-lasting even after the emergency has ended, with the risk of developing into chronic distress and psychopathological conditions (e.g., [9,10]). To our knowledge no study focused on the relationship between unemployment as a consequence of the pandemic and QoL as a global health indicator, within the Italian context. A recent study by Medda and colleagues [11] found that the COVID-19 pandemic was associated with an increased depressive symptomatology by the end of the first lockdown measures particularly for women, young people and individuals living in a pre-existing economically disadvantaged environment. In another investigation with a general Italian population sample, Rossi and colleagues [12] found that being unemployed and having a discontinued working activity during the pandemic were associated with poorer mental health. In both studies it is unclear if economic problems as well as unemployment were either antecedents to or consequences of the COVID-19 outbreak. To our knowledge, only one study conducted in Italy considered job loss as a consequence of the pandemic [13]. The authors found a relationship between being unemployed because of the pandemic and both poorer general mental health and higher depressive symptoms. However, so far no investigation focused on the underlying mechanisms that can help us to better understand the psychological processes through which job loss during the pandemic impacts individuals’ health and wellbeing.

The negative impact on employment caused by the pandemic may have attacked confidence in the future, which is a crucial component of individual well-being, thus generating a sense of hopelessness, namely the perception that the future will have negative outcomes. Precisely, hopelessness is a psychological construct that refers to a system of negative attitudes toward the future [14], and correlates with many psychosocial stressors, including low levels of education, unemployment, financial difficulties [15], and lower QoL [16]. Particularly, socioeconomic factors, such as low employment rates [17], are among the contextual determinants of hopelessness, while positive emotional climates and relationships seem to play a protective role against it. In light of all this, it is easy to imagine that a stressful episode such as job loss during the pandemic could increase a sense of hopelessness, and that this can in turn affect important outcomes for individual wellbeing and health.

To better understand the processes at work behind the impact of stressful events on people’s health, it is essential to examine the buffering potential of individual’s psychological resources [18]. The stress-buffering model proposes that personal assets can attenuate the negative effects of risk factors over a person’s health [19,20]. According to this model, the adverse impact of environmental risks will be weaker for individuals who have higher levels of personal resources. In the case of current pandemic, people are facing with chronic daily stress (the pandemic itself), as well as with its possible consequences which constitute additional external stressors (e.g., daily life restrictions, loss of a beloved person and job loss). However, although it is well proven that both these categories of stressors affect mental health and well-being [21,22,23], it is also known that not all people who face stressful circumstances experience impairments in their daily lives and psychological wellbeing. The strength of the association between stress and psychological responses actually depends on individual differences, in terms of personal characteristics and strategies [24,25]. Nevertheless, assumptions that merely include direct effects of life-stressors on health as well as of psychological variables on health, without considering specific external circumstances, are incomplete and generally neglect intervening factors, leading to a potentially inaccurate estimation of effect sizes.

Emotional Intelligence (EI) is recognized by the international literature as a factor that promotes general health [26,27] and better adaptation [28], and which influences how individuals cope with demands and pressures from the environment to be more resilient to challenging situations [29]. Particularly, the construct of trait EI has emerged as an important individual difference variable referring to a constellation of emotional self-perceptions assessed by self-reported questionnaires [30]. Essentially, trait EI concerns people’s perceptions of their emotional dispositions and is identified as a distinct latent variable which integrates the affective aspects of personality [31,32]. Several studies have explored the influence of trait EI across the lifespan (see [33] for a review) and its impact on health, underlining that high trait EI strongly positively predicts well-being and mental health (e.g., [34]), and moderates responses to stress [35].

Overall, the protective effect of trait EI has been recently confirmed in highly stressful scenarios such as those connected with the current COVID-19 pandemic: for instance, high levels of trait EI have been shown to predict a lower intensity of negative emotions during pandemic [36], lower prevalence of burnout, psychosomatic complaints, and a favourable effect on job satisfaction in healthcare workers [37], as well as a use of better coping strategies [38]. Moreover, findings suggest that trait EI may account for individuals’ effectiveness in managing their emotions and obtaining social support which, in turn, may ameliorate their worries about COVID-19 [39].

### The Present Study

Studies performed in the Italian context [3,11,12,13,40,41] as well as in other countries [42,43] found that pre-existing disadvantageous socioeconomic positions can be significant risk factors for a poorer mental health during the pandemic period. Few studies focused instead on protective factors, such as social support [44]. To our knowledge there are no published studies where real-life risk factors and protective individual differences variables are simultaneously taken into consideration to better understand the impact of the pandemic over health.

In light of the foregoing and considering that there is a need to focus on the identification of resources as well as risk factors for psychological health during adversity (e.g., [45]), the present study aims to provide further country-specific evidence on the extent to which the pandemic and its consequences affected individuals’ health operationalized as QoL. Particularly, in the current research we propose a theoretical model that examines whether job loss during the COVID-19 pandemic and hopelessness were associated with QoL amongst a representative sample of Italian workers. We also examined whether this relationship would be moderated by the protective role of trait EI. In particular, with the aim of filling the existing gap in the literature we tested a moderated mediation model, which is shown in Figure 1. Therefore, the main goal of the present study was to test a model testing the roles of both protective (trait EI) and risk (hopelessness) internal psychological factors on the impact of an external stressor (job loss) over QoL.

On the basis of current literature, several hypotheses were tested and we anticipated the following expected results: (1) individuals who had significant changes in employment due to the COVID-19 pandemic would have higher levels of hopelessness and lower levels of QoL; (2) QoL will negatively correlate with hopelessness and positively with QoL; (3) hopelessness will mediate the relationship between job loss and QoL; (4) trait EI will moderate the direct and/or indirect pathways from job loss to QoL. Given that trait EI is conceptualized as a personality dimension, no hypothesis on the differences between workers depending on changes in employment was formulated.

## 2. Materials and Methods

### 2.1. Participants

Inclusion criteria for sample selection were age greater than 18 years, having a paid job before the coronavirus outbreak, and living in Italy at the time of quarantine. Of the overall number of respondents (*N* = 2332), *n* = 722 were excluded (*n* = 455 university students, *n* = 140 retired, *n* = 117 missing data, and *n* = 10 resident outside Italy). Our final sample included 1610 Italian workers (women *n* = 1172, 72.8%). Of these, 439 (27.3%) reported to have lost their job due to the pandemic. Prior the pandemic, 1046 respondents worked as employees (e.g., public administration and private companies), while the remaining 564 were self-employed. Respondents were middle-aged adults (age range 35-64; *n* = 1018, 63.2%), young adults (age range 18–34; *n* = 534, 33.2%) and few older adults (age range ≥ 65; *n* = 58, 3.6%). Most participants had a university degree (*n* = 971; 60.3%), and were in a stable relationship (*n* = 797; 49.5%). Data came mainly from the South (*n* = 798; 49.6%) and North (*n* = 693; 43%) regions of Italy, with the central regions being less represented (*n* = 80; 5%). Thirty-nine (2.4%) participants did not report information on their residence. At the time of data collection none of the participants had contracted the coronavirus nor had lost a friend or a relative because of it.

### 2.2. Instruments

Demographics were collected through an ad hoc questionnaire which included items on sex, age, marital status, education level, occupational status, region of residence in Italy, presence of medical and psychiatric diagnosis, and job loss due to the pandemic.

The World Health Organization Quality of Life BREF Assessment Instrument (WHOQOL-BREF) was used to assess QoL [46,47]. The WHOQOL-BREF is a 26 items version of the WHOQOL-100. Items ask respondents to rate their QoL during the last two weeks on a 5-point Likert scale. For the purposes of the present study, only the global QoL score was used (Cronbach’s α=0.88).

The Trait Emotional Intelligence Questionnaire–Short Form (TEIQue-SF; [48]) was employed to measure trait EI. Through 30 items rated on a 7-point Likert scale, the TEIQue provides a global trait EI score and scores at four factors (i.e., wellbeing, self-control, emotionality and sociability) that correlate meaningfully with a wide range of diverse criteria pertaining the domains of health, clinical, educational and occupational psychology [31] Andrei et al., 2014). For the purposes of the present study, only the global trait EI composite score was used, showing a good reliability (Cronbach’s α=0.88).

The Beck Hopelessness Scale (BHS; [14]) is a self-report instrument for the quantification of hopelessness in non-psychiatric, as well as psychiatric patients. BHS includes 20 true/false items about the future (9 positive and 11 negative), that can be summed up to give a total score ranging from 0 to 20. Higher scores reflect increased hopelessness. Reliability for the present study was good (Cronbach’s α=0.86).

### 2.3. Procedure

An online cross-sectional data collection was performed with Qualtrics^®^ Survey Platform (Qualtrics: Seattle, USA). This data collection strategy was chosen as it allowed us to reach as many participants as possible in a phase of forced social distancing. Participation was voluntary with no incentives. Data collection started after 7 weeks of quarantine in Italy (25 April 2020) and was performed for about 6 weeks, until the end of lockdown measures (2 June 2020). The study was approved by the Bioethics Committee of the University of Palermo (n. 4/2020). A more detailed description of study procedure can be found in a recent article by Epifanio and colleagues [3].

### 2.4. Statistical Analysis

Statistical analyses were performed using SPSS (version 25) for Windows [49] (IBM Corporation, NY, USA). Pearson’s correlations were used to investigate associations among variables. The computational tool for SPSS, PROCESS (NY, USA) [50], was used to test our models. PROCESS automatically generates bootstrap confidence intervals in order to control for the possible non-normality of the sampling distribution. If compared to other models, it provides a greater balance between Type I error and power and generates the most accurate confidence intervals for indirect effects [50,51].

In the case of the present study, the meditation model was tested as a first step (Model 4 from PROCESS) and the mediating role of hopelessness from job loss to QoL was examined. Job loss was coded as a dummy variable, with 0 being the condition with no significant job changes due to the pandemic and 1 being the condition where workers lost their job because of the pandemic. Mediation is supported when the confidence intervals do not contain zero. The moderated mediation model was tested as a second step (Model 59 from PROCESS). The potential moderating role of trait EI in the previously established mediation was examined. Moderated mediation addresses the interaction between job loss and trait EI over QoL (the residual direct relationship), the interaction between job loss and trait EI over hopelessness (the first part of the mediation process), and the interaction between hopelessness and trait EI over QoL (the second part of the mediation process). Continuous variables were mean centered. Interaction effects can be identified when the confidence intervals do not contain zero. In these analyses, gender and age range were controlled by entering them into regression equations as covariates. Subsequently they did not result underlying factors explaining the direct and indirect associations of job loss with QoL [52].

## 3. Results

### 3.1. Testing for Hypothesis 1 and 2: Preliminary Analysis

Table 1 presents Pearson’s correlation coefficients of study variables. Specifically, both job loss (*r* = −0.09, *p* < 0.001) and hopelessness (*r* = −0.38, p < 0.001) were negatively associated with QoL, whereas trait EI (*r* = 0.39, *p* < 0.001) showed a positive correlation with QoL. In addition, job loss was negatively related to trait EI (*r* = −0.12, *p* < 0.001) and positively to hopelessness (*r* = 0.12, *p* < 0.001). Finally, trait EI and hopelessness were negatively correlated (*r* = −0.58, *p* < 0.001).

### 3.2. Testing for Hypothesis 3: Mediating Role of Hopelessness

The aim of Hypothesis 3 was to examine the potential mediating role of hopelessness in the relationship between job loss and QoL during the COVID-19 pandemic. After controlling for the covariates, hopelessness partially mediated the relationship between job loss and QoL (B = −0.81, 95% confidence interval = [−1.18, −0.48]). Specifically, job loss positively predicted hopelessness (B = 1.24, *p* < 0.001), which in turn negatively predicted QoL (B = −0.66, *p* < 0.001). The negative direct association between job loss and QoL remained significant (B = −1.63, *p* < 0.001).

### 3.3. Testing for Hypothesis 4: Moderating Role of Trait EI

Hypothesis 4 referred to the examination of trait EI as potential moderator in the direct and/or indirect relationship between job loss and QoL during the COVID-19 pandemic via hopelessness. Moderated mediation model was tested with PROCESS (Model 59) to obtain bootstrapped confidence intervals for mediation model at different levels of trait EI. In particular, the parameters for two regression models were estimated. Model 1 estimated the moderating role of trait EI in the relationship between job loss and hopelessness. Model 2 estimated the moderating role of trait EI in the relationship between hopelessness and QoL, as well as the residual direct relationship between job loss and QoL. The effects of gender and age were controlled as these variables were included in the models as covariates. Table 2 presents the results of the two models.

After controlling for the covariates, Model 1 indicated that job loss significantly and positively predicted hopelessness (B = 0.63, *p* < 0.01), while its interaction with trait EI in predicting hopelessness was non-significant (B = −0.12, *p* = 0.66, 95% confidence interval = [−0.67,.43]). The full model accounted for 35% of the variance in hopelessness (R^2^ = 0.35, *p* < 0.001).

Model 2 indicated that hopelessness significantly and negatively predicted QoL (B= −0.43, *p* < 0.001), and this relationship was moderated by trait EI (B = −0.11, *p* < 0.05, 95% confidence interval = [−0.22, −0.05]). The simple slope effect also indicated that the indirect effect of job loss on QoL through hopelessness was observed when trait EI was high, moderate and low, as shown in Figure 2.

The simple slope for high (+1 *SD*), moderate, and low (−1 *SD*) levels of trait EI were (B = −0.51, *p* < 0.001, 95% confidence interval = [−0.65, −0.36]), (B = −0.43, *p* < 0.001, 95% confidence interval = [−0.53, −0.33]), and (B = −0.35, *p* < 0.001, 95% confidence interval = [−0.45, −0.25]), respectively.

In addition, job loss and trait EI interacted to predict QoL (B = 1.98, *p* < 0.001, 95% confidence interval = [.87, 3.10]), while the direct effect of job loss over QoL became non-significant (B = −0.55, *p* = 0.16, 95% confidence interval = [−1.32, 0.22]). As Figure 3 shows, The adverse impact of job loss on QoL levels was stronger for workers who had low levels of trait EI (B = −0.35, *p* < 0.001, 95% confidence interval = [−0.45, −0.25]). The full model accounted for 20% of the variance in QoL (R^2^ = 0.20, *p* < 0.001).

## 4. Discussion

Although the literature suggests that stressful events such as job loss significantly impact important health-related variables such as QoL, the underlying mediation and moderation dynamics are less clear. To explore such dynamics more in depth, the present study constructed a moderated mediation model. To summarize, our first three hypotheses pertaining the associations among the studied variables were supported, and hopelessness mediated the relationship between job loss and QoL levels. However, a full mediation was not obtained as further risk factors, including alienation [53], and/or resources, such as self-efficacy and social support [25,54], could play a significant role in the buffering process and could potentially contribute to mediate the association between job loss and QoL.

The fourth hypothesis of the present study was partially supported as we found that trait EI moderated the second part of the mediation process, namely the effect of hopelessness on QoL, and the residual direct association between job loss and QoL. Nevertheless, trait EI did not moderate the first part of the mediation process (the effect of job loss on hopelessness levels). This finding suggests that trait EI does not serve as a buffer against the negative role played by job loss on hopelessness levels, therefore, job loss remains a salient risk factor for an increase in hopelessness. This finding is in line with the literature on the association between unemployment and job-loss with poorer mental health and higher depression, which occurs during the pandemic as well [12,13]. At the same time, high trait EI acts as protective factor over QoL, mitigating the impact of both job loss and hopelessness over QoL levels during the pandemic. This finding highlights the importance of individual resources not only after a critical life event (i.e., job loss) but also and especially during a time of major crisis, where social distancing and the quarantine measures put individuals’ QoL to the test even further [3].

Although there is strong evidence for the associations of hopelessness and trait EI with health outcomes (e.g., [34,55]), their combined roles in the relationship between a major critical life event (i.e., job loss) and QoL remained unexplored until now. Our results suggest a need to consider trait EI as a potential psychological resource that might mitigate the impact of stressful life events and subsequent negative cognitive and emotional reactions, such as hopelessness, over QoL. Particularly, present findings highlight that low trait EI levels may be particularly detrimental when external stressors occur, as the adverse impact of job loss on QoL levels was stronger for workers who had low levels of trait EI. Our study supports the need to integrate considerations on both psychological risk and protective factors into COVID-19 care, including the monitoring of psychological symptoms and social needs within the general population. Future studies could examine whether it is possible to test for a cumulative protective factor index that incorporates protective factors from multiple different domains but having similar buffering properties against the adverse effects of the pandemic. This line of research may act as a critical step to develop effective intervention strategies and provide mental health services during and after current sanitary emergency.

The negative conditions related to the pandemic, including job loss, have the potential to contribute to what is known as pandemic fatigue, a latent phenomenon emerging gradually over time and affected by emotions, experiences and perceptions, defined as distress which can result in demotivation to follow recommended health protective behaviours [56]. Indeed, pandemic fatigue has been recognized as a predictor of nonadherence to health protective measures and impact of noncompliance with public health measures on COVID-19 disease control [57], therefore it poses a serious threat to national and global efforts by governments and health-authorities worldwide to support pandemic prevention and management and to control the spread of the virus via both recommended and mandated protective measures (i.e., physical distancing, mask wearing, and social-isolation). For this reason, the identification of psychological variables that may serve as protective/promoting factors and/or act as risk factors for a better QoL is crucial for the development of interventions aimed at reducing the emotional impact of the pandemic on people’s psychological and physical health. Consequently, reducing individual psychological vulnerability may support people in maintaining and strengthening preventive health measures to cooperate in reducing the spread of infection.

This study had several limitations that should be considered when interpreting our results. These include the cross-sectional design, the higher prevalence of female participants, and the use of self-reports only which may be associated with common method bias. Additionally, although recruitment procedures (i.e., snowball sampling method through social media, emails and university’s website) allowed us to reach as many voluntary participants as possible during forced social distancing, they may have biased sample’s composition in several ways. For instance, sample selection was strongly related to Universities’ geographical location, thus impacting the representativeness of Italian population as the central regions of the country had a low participation rate. Moreover, online recruitment procedures may naturally select individuals who are more active on both the internet and social media platforms, thus neglecting potential individual differences in how people deal with stress. On the one hand, individuals who are active on social media can feel greater social support in times of social distancing thanks to facilitated interpersonal connections [58]. However, on the other hand, recent evidence suggests that active social media use during the pandemic can leave people be more susceptible to greater stress and depression [59]. All this may in turn impact QoL levels and psychological symptoms, including hopelessness. These elements should be carefully taken into consideration by future studies, together with the use of random sampling procedures and a more heterogeneous sample selection, particularly in terms of gender.

Moreover, further variables, such as the income sources of the participants and the type of work they had before the pandemic, which may potentially mediate and/or moderate the relationship between job loss and QoL should be considered by future research. Last, new research may focus on families rather than singular citizens in order to have a broader picture of the effects of job loss on QoL. Variables such as family relations among participants, number of members in a family who lost the job, and family size are all important factors which may potentially mediate or moderate the outcomes/effect.

## 5. Conclusions

Although much attention has been paid to the effects of the COVID-19 pandemic over health-related variables, little has been carried out so far to understand the underlying specific mediation and moderation mechanisms. The present study is the first to examine the role of hopelessness and trait EI in the impact of job loss on QoL during the COVID-19 pandemic. The model that was tested herein may more effectively reflect the complexity of the conditions through which a significant specific stressful event subsequent to the pandemic emergency affects individuals’ QoL. Indeed, the model included both protective (trait EI) and risk (hopelessness) psychological factors which may have influenced job loss’ impact over QoL. Our findings indicated that the relationship between job loss during the pandemic and QoL was partially mediated by hopelessness and moderated by trait EI and these effects were gender- and age range- independent.

This study identifies a pathway (i.e., via hopelessness) that may help to better understand the mechanism by which job loss affects individual’s QoL. Moreover, the findings suggest that both the direct and indirect job loss/QoL associations vary with levels of trait EI. Such results indicate that both contextual (job loss) and individual factors (hopelessness levels and trait EI) that may impact QoL levels, especially during emergency times, are interrelated, namely, not independent. Therefore, we believe that health promotion policies must consider and address both contextual and personal variables, so that intervention for high-risk group can be planned in advance.

## Figures and Tables

**Figure 1 ijerph-19-02756-f001:**
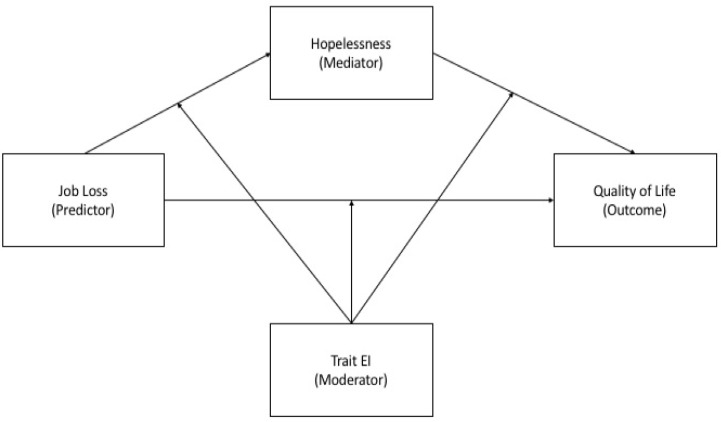
Moderated mediation model of associations between job loss, trait EI, hopelessness and quality of life.

**Figure 2 ijerph-19-02756-f002:**
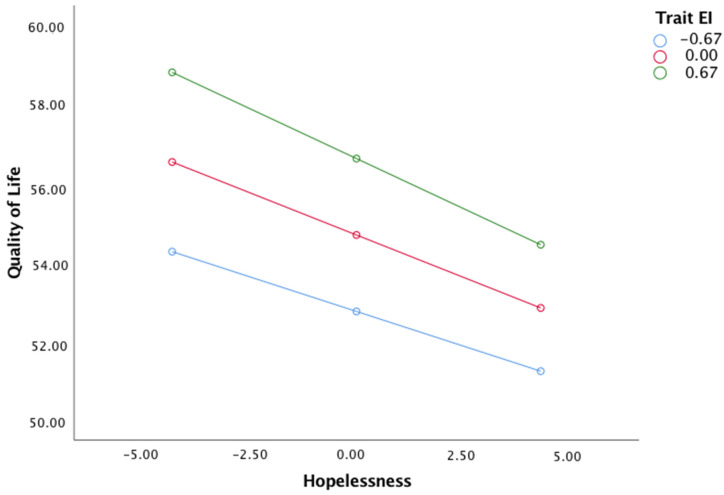
Data plot of simple slope interaction for hopelessness × trait EI on Quality of Life.

**Figure 3 ijerph-19-02756-f003:**
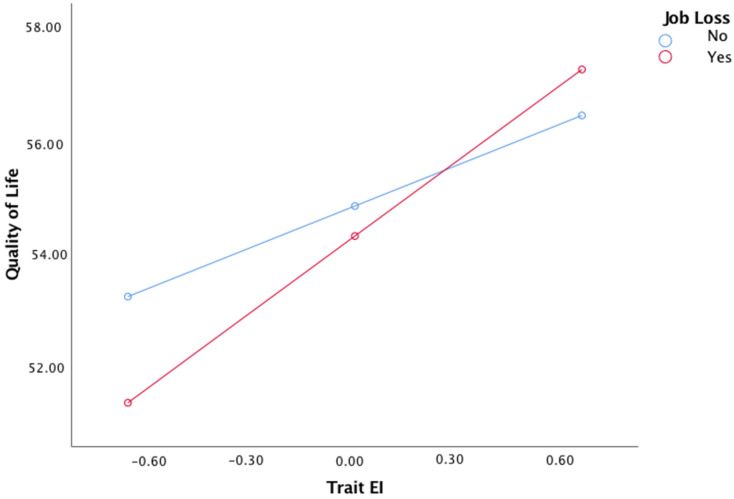
Data plot of simple slope interaction for job loss × trait EI on Quality of Life.

**Table 1 ijerph-19-02756-t001:** Correlations for all study variables.

Variables	Gender	Age	Job Loss	QoL	Hopelessness	Trait EI
Gender	.					
Age	0.03	.				
Job loss	0.00	−0.14 **	.			
QoL	0.07 **	0.00	−0.09 **	.		
Hopelessness	−0.01	0.04	0.12 **	−0.38 **	.	
Trait EI	0.01	0.09 **	−0.12 **	0.39 **	−0.58 **	.

Note. Gender and Job loss were dummy coded such that 0 = male and 1 = female, and 0 = No changes in employment during the pandemic and 1 = Job loss because of the pandemic, respectively. QoL = Quality of life; Trait EI = Trait emotional intelligence. ** *p* < 0.01.

**Table 2 ijerph-19-02756-t002:** Testing the moderated mediation effects of changes in employment on quality of life during the COVID-19 pandemic.

Model	Predictors	*B*	*SE*	*t*	95% CI
Model 1 (Hopelessness)	Gender	−0.10	0.20	−0.50	[−0.48; 0.28]
	Age	0.86 ***	0.17	5.11	[0.53; 1.19]
	Changes in Employment	0.63 **	0.20	3.16	[0.24; 1.03]
	Trait EI	−3.75 ***	0.16	−23.17	[−4.07; −3.03]
	Changes in Employment × trait EI	−0.12	0.28	−0.43	[−0.67; 0.43]
	R^2^	0.35			
	F	172.63 ***			

Model 2 (QoL)	Gender	1.10 **	0.38	2.86	[0.35; 1.86]
	Age	−0.24	0.33	−0.71	[−0.89; 0.41]
	Changes in Employment	−0.55	0.39	−1.39	[−1.32; 0.22]
	Hopelessness	−0.42 ***	0.05	−8.17	[−0.52; −0.32]
	Trait EI	2.41 ***	0.36	6.57	[1.69; 3.12]
	Changes in Employment × trait EI	1.98 ***	0.57	3.49	[0.87; 3.10]
	Hopelessness × trait EI	−0.11 *	0.05	−2.05	[−0.22; −0.05]
	R^2^	0.20			
	F	58.01 ***			

Note. Gender and Job loss were dummy coded such that 0 = male and 1 = female, and 0 = No changes in employment during the pandemic and 1 = Job loss because of the pandemic, respectively. Trait EI = Trait emotional intelligence; QoL = Quality of life. * *p* < 0.05 ** *p* < 0.01 *** *p* < 0.001.

## Data Availability

Data are available upon request due to privacy restrictions.

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
