# Peer review of "Quality of Life and Job Loss during the COVID-19 Pandemic: Mediation by Hopelessness and Moderation by Trait Emotional Intelligence"

_ijerph, 2022, doi:10.3390/ijerph19052756_

Round 1

Reviewer 1 Report

The authors have addressed my comments in the discussion section. I have no further questions.

Reviewer 2 Report

The  authors have addreesed all comments and I endorse publication

This manuscript is a resubmission of an earlier submission. The following is a list of the peer review reports and author responses from that submission.

Round 1

Reviewer 1 Report

The results about the role of hopelessness and trait EI in the impact of job loss on QoL during the COVID-19 pandemic are very importante to help patients and health workers. 

Reviewer 2 Report

There are several crucial factors that the authors did not consider in the study design and these have to be addressed:

  1. Since the survey was conducted through email and other social media, how many of the participants were coming from the same family, or were relatives?
  2. Number of members in a family who lost the jobs, the income sources of the participants, the type of works they had before the pandemic and the size of the family etc. are all important factors potentially could mediate or moderate the outcomes/effect. These factors need to be built into the model.
  3. The authors should discuss the possible bias of the study associated with how participants were recruited (i.e. through social media, emails and university’s website). For example, the way to deal with stress for the participants who are active on social media (with social support) could be very different compared to those who are not.
  4. No information was provided on how many surveys were taken and what were the inclusive and exclusive criteria. Did the study provide any incentives to recruit the subjects?
  5. It is well known that job lost will affect the quality of life and personal EI is protective against stress in normal situation even without the pandemic. The biggest concern this reviewer with the study therefore is that the authors fail to investigate how unique the pandemic contributes to the relationship/equation compared to the job lost otherwise during a normal scenario.     
  6. Proof reading is needed, for example: Line 41 “(e.g., [1, 2, 3, 4].” Is missing half of the parenthesis. Line 125: QoL will…corelate…positively with QoL;

Reviewer 3 Report

 This paper investigates the effects of the COVID-19 pandemic by examining a moderated mediation model in which the impact of job loss over quality of life (QoL) is mediated by hopelessness and moderated by trait emotional intelligence (trait EI). Participants were adult workers (N = 1610), who completed a series of anonymous online questionnaires. Total, direct and indirect effects were estimated with bootstrapped mediated moderation analyses, where the effects of gender and age range were controlled. In this model Job loss was found to be negatively associated with QoL, and hopelessness partially mediated such relationship. These relationships were in turn moderated by trait EI. Our study suggests that trait EI levels act as protective factor for a good QoL, mitigating the impact of both job loss and hopelessness. The finding are useful because identifying psychological protective and/or risk factors for a better QoL helps the development of interventions for reducing the negative real-life consequences.

This research is well perfumed using mediation and moderation regression models.

Valid scales were implemented to measure the variables involved, and for statistical calculations the PROCESS program (SPSS) was used, which a good choice for such analysis. The methodological part is well presented and comprehensible to the reader.

Overall the paper is characterized by a completeness as far as the methodological part and articulacy as far as the presentation of the theoretical and conclusion parts.

Please find the attachment. I endorse publications after minor revision. 
